# Quality of Experience (QoE) in Cloud Gaming: A Comparative Analysis of Deep Learning Techniques via Facial Emotions in a Virtual Reality Environment

**DOI:** 10.3390/s25051594

**Published:** 2025-03-05

**Authors:** Awais Khan Jumani, Jinglun Shi, Asif Ali Laghari, Muhammad Ahmad Amin, Aftab ul Nabi, Kamlesh Narwani, Yi Zhang

**Affiliations:** 1School of Electronic and Information Engineering, South China University of Technology, Guangzhou 510641, China; eeawais.jumani@mail.scut.edu.cn (A.K.J.); eeahmadamin@mail.scut.edu.cn (M.A.A.); eeaftab.shahani@mail.scut.edu.cn (A.u.N.); kdnarwani@hotmail.com (K.N.); yzhang817@163.com (Y.Z.); 2Software College, Shenyang Normal University, Shenyang 110136, China; asiflaghari@synu.edu.cn

**Keywords:** deep learning, cloud gaming, virtual reality, quality of experience, model assessment, emotion analysis

## Abstract

Cloud gaming has rapidly transformed the gaming industry, allowing users to play games on demand from anywhere without the need for powerful hardware. Cloud service providers are striving to enhance user Quality of Experience (QoE) using traditional assessment methods. However, these traditional methods often fail to capture the actual user QoE because some users are not serious about providing feedback regarding cloud services. Additionally, some players, even after receiving services as per the Service Level Agreement (SLA), claim that they are not receiving services as promised. This poses a significant challenge for cloud service providers in accurately identifying QoE and improving actual services. In this paper, we have compared our previous proposed novel technique that utilizes a deep learning (DL) model to assess QoE through players’ facial expressions during cloud gaming sessions in a virtual reality (VR) environment. The EmotionNET model technique is based on a convolutional neural network (CNN) architecture. Later, we have compared the EmotionNET technique with three other DL techniques, namely ConvoNEXT, EfficientNET, and Vision Transformer (ViT). We trained the EmotionNET, ConvoNEXT, EfficientNET, and ViT model techniques on our custom-developed dataset, achieving 98.9% training accuracy and 87.8% validation accuracy with the EmotionNET model technique. Based on the training and comparison results, it is evident that the EmotionNET model technique predicts and performs better than the other model techniques. At the end, we have compared the EmotionNET results on two network (WiFi and mobile data) datasets. Our findings indicate that facial expressions are strongly correlated with QoE.

## 1. Introduction

Cloud gaming, also known as gaming on demand or Gaming as a Service (GaaS), represents a modern way of gaming where service providers harness the power of cloud servers to run games and stream high-quality visual content and audio directly to players’ devices [1,2]. With the evolution of cloud gaming, there is no longer a need for expensive personal gaming hardware [3,4]. Players can select and play any game virtually on cloud servers, provided they have a stable internet connection [5,6]. This is particularly beneficial for a new generation of gamers who cannot afford high-end devices, as their local devices may not support high-graphics games [7,8]. Cloud gaming makes it possible to play high-graphics games on local devices [9,10]. One of the key benefits of cloud gaming is its ability to continuously upgrade with the latest technologies, providing users with an immersive gaming experience [11,12]. However, cloud gaming is highly dependent on a good internet connection, as players access games remotely from servers. Low latency is crucial for a responsive and enjoyable user experience [13]. High-bandwidth internet connections, such as fiber-optic internet and stable 5G networks, are required for playing high-graphics games [14]. Cloud gaming data centers need powerful hardware because these servers run games through virtual machines, requiring significant processing power, graphics processing units (GPUs), and ample storage capacity [15]. As the demand for cloud gaming grows, there is a need for advanced servers that can run games smoothly [16,17]. Content Delivery Networks (CDNs) play an essential role in cloud gaming by delivering game data smoothly and reliably from the nearest server to the user’s location [18,19]. An efficient CDN infrastructure is key to providing a lag-free cloud gaming experience, especially for players around the world [20]. In terms of the latest technological advancements, artificial intelligence (AI) is becoming increasingly important for cloud gaming, as it can enhance the overall experience. AI can be used for dynamic bitrate streaming, adjusting video quality based on the user’s internet connection to minimize buffering and improve performance [21,22]. Additionally, AI can personalize gaming settings and difficulty levels based on the player’s behavior, leading to a more tailored gaming experience. Since COVID-19, cloud gaming has become more accessible and in demand, making high-performance gaming available to everyone, regardless of their device specifications or location [23,24]. While cloud gaming offers many advantages, there are still some limitations, such as latency, bandwidth, accessibility, and availability [25]. Cloud infrastructure depends heavily on high-speed internet connections, which can affect the user’s QoE.

Several studies have explored cloud gaming QoE using deep learning-based techniques. For instance, Barman [26] focused on objective QoE evaluation based on network packet loss, yet this method does not consider real-time user emotions, which are crucial for QoE assessment. Also, cox [27] proposed an edge-based streaming solution but did not assess player satisfaction directly, limiting its applicability in user experience analysis. Likewise, li [28] introduced a video quality assessment model, but it was not specifically designed for cloud gaming, making it less effective in dynamic gaming scenarios. Physiological sensor-based approaches, such as EEG-based emotion recognition, provide objective indicators of user engagement but introduce challenges related to hardware costs, user discomfort, and scalability [29]. Other works have leveraged heart rate variability (HRV) and skin conductance; however, these methods require additional sensors, which may not be practical for widespread deployment in cloud gaming [30]. Deep learning-based methods for video classification and QoE assessment have demonstrated high accuracy, particularly CNN-based approaches. However, existing studies primarily rely on network traffic analysis and video quality metrics, failing to incorporate real-time player emotions. Moreover, conventional QoE models have typically emphasized Quality of Service (QoS) parameters, such as latency and bandwidth, rather than subjective user experiences. These limitations highlight the need for a real-time, emotion-driven QoE assessment technique. Unlike prior studies, our proposed EmotionNET model integrates facial emotion recognition with deep learning to provide a more accurate and dynamic assessment of QoE in cloud gaming environments.

Similarly, Chen [31] focused on the classification of E-sports live streaming videos by leveraging deep learning techniques, particularly CNNs. This research highlights the importance of high-quality video in enhancing users’ perceptions and experiences during E-sports streaming. Traditional methods for video classification often rely on network traffic analysis and the statistical characteristics of video flows. The study compared various deep learning classifiers, demonstrating that CNNs, combined with specific hyper-parameters, achieve the highest accuracy in multi-class classification tasks, reaching up to 97%. The research emphasized the significance of selecting appropriate traffic intensity features and optimizing hyper-parameters to improve the classification performance. This study contributes to the field by providing a robust method for E-sports operators to classify and improve the quality of live streaming videos, thereby offering differentiated services to their users.

Also, Shang [32] focused on the challenges and methodologies for assessing the quality of high-motion live-streaming videos, such as those used for sports events. The paper highlights the development of the LIVE Livestream Database, which includes 315 videos of high-motion sports content, impaired by six common types of distortions. A subjective quality study involving over 12,000 human opinions from 40 subjects was conducted to gather Mean Opinion Scores (MOSs). This database and study aimed to facilitate the development, testing, and comparison of objective VQA algorithms. The authors emphasized the need for such databases due to the unique challenges posed by high-motion content, including motion blur and stutter, and the limitations of existing VQA databases that typically do not address these issues comprehensively. The new database is intended to advance research in predicting the perceptual quality of high-motion, live-streamed videos and is publicly available for research purposes. Furthermore, Zhang [33] provided a comprehensive analysis of QoE models for online video streaming, emphasizing the importance of subjective and user-oriented assessments over traditional Quality of Service (QoS) metrics. The authors categorized the development of QoE models into four stages: QoS monitoring, subjective tests, objective quality models, and data-driven quality models. They discussed the evolution from early handcrafted methods to advanced learning-based models, highlighting the shift towards integrating Human Visual System (HVS) characteristics and leveraging large-scale data for more accurate QoE assessments. The paper particularly focused on objective and data-driven models, noting the advantages of each approach in balancing computational efficiency and estimation accuracy. This work underscored the need for accurate QoE models to enhance user satisfaction in the rapidly growing online video streaming market.

Despite advancements in QoE assessment for cloud gaming, existing studies have several limitations. Many approaches primarily rely on network-based QoS metrics or subjective user surveys, which do not accurately capture real-time user experiences. Some studies focus on network conditions such as packet loss and latency but fail to incorporate emotional responses, which are crucial for assessing true user satisfaction. Others propose streaming optimization techniques but lack direct player experience evaluation. Additionally, video quality assessment models designed for streaming services are often applied to cloud gaming without considering its dynamic and interactive nature. While physiological sensor-based methods such as EEG [34] and heart rate variability have been explored, they require additional hardware, making them impractical for large-scale deployment. Furthermore, many studies rely on small or constrained datasets, limiting the generalizability of their findings. Traditional deep learning-based QoE models primarily emphasize video quality or network performance, often overlooking the role of user emotions during gameplay. To address these gaps, this study introduces EmotionNET, a deep learning-based framework that leverages facial emotion recognition for real-time QoE assessment in cloud gaming. Unlike prior methods, EmotionNET directly analyzes players’ emotional responses, providing a more accurate and dynamic evaluation of their gaming experience. Additionally, its performance is validated against state-of-the-art deep learning models (ConvoNEXT, EfficientNET, and ViT), ensuring a robust and scalable approach to QoE assessment.

Table 1 provides a comprehensive comparison of several object detection algorithms, highlighting key attributes such as model architecture, accuracy, speed, training datasets, input size, use of pre-trained models, and implementation complexity. It starts with the DCM3-YOLOv4 algorithm, which utilizes a CNN architecture and achieves a high accuracy of 94.3%. It was trained on datasets derived from MAFA and WIDER FACE, with an input size of 4000. However, it does not use pre-trained models, and its implementation is moderately complex. Likewise, the SSD algorithm stands out for its use of the ResNet v2 and VGG16 architectures, achieving exceptionally high accuracy for face recognition tasks. It performs well even in occluded conditions, with a 95.38% accuracy at a 50% occlusion rate from one meter away. SSD was trained on the extensive MS1M-ArcFace and CelebA datasets, with input data ranging from 85,000 to 5.8 million face images. It also operates at 8 frames per second (FPS) and uses pre-trained models, making it moderately complex to implement. Also, EfficientDet, which employs a BiFPN architecture, achieved an accuracy of 81.74%. It was trained on the COCO 2017 dataset with an input size of 118K images. It also leverages pre-trained models and has a moderate implementation complexity. Moreover, CenterNet, built on the CornerNet architecture, has an accuracy of 84.5%. It uses the MS COCO dataset with an input size of 511 × 511. This model does not use pre-trained models and is moderately complex to implement. Finally, Cascade R-CNN, using a Parallel Cascade R-CNN architecture, has an accuracy of 78.96%. It was trained on the DOTA dataset, which is designed for object detection in aerial images, with input sizes ranging from 800 × 800 to 4000 × 4000. It uses pre-trained models and is moderately complex to implement. Furthermore, each model detects facial emotions which are extracted from videos as 25 fps and images of size 48 × 48. The EmotionNET model achieved an accuracy of 98.9%, with an easy complexity level. ConvoNEXT (ResNet) performed at 94.9% accuracy but it had an overfitting problem, and it had moderate complexity. EfficientNET-B0 achieved a 92% accuracy and was also categorized under moderate complexity. Lastly, the Vision Transformer (ViT) model recorded 91% accuracy and was rated as having moderate complexity as well. There is no previous work related to facial emotion detection using deep learning techniques. We have applied the three more well-known deep learning techniques to check our EmotionNET technique robustness. More justification and proof is discussed in the Section 3.

Despite extensive research on QoE assessment in cloud gaming, existing methods remain limited in their ability to capture real-time, emotion-driven user experiences. Traditional QoS-based approaches fail to account for subjective perception, while physiological sensor-based methods introduce practical constraints due to additional hardware requirements. Deep learning techniques have been explored for video quality and network-based assessments, but their direct application to emotion-driven QoE evaluation remains underdeveloped. Moreover, previous studies often focus on static datasets rather than real-time gameplay conditions, making them less applicable to dynamic cloud gaming environments. This study bridges these gaps by introducing EmotionNET, a CNN-based model designed to assess QoE directly through facial emotion recognition in real-time gameplay, eliminating the need for additional sensors or subjective surveys. Our comparative evaluation with ConvoNEXT, EfficientNET, and ViT further validates its effectiveness, demonstrating its potential as a scalable and robust approach for QoE assessment in cloud gaming.

The cloud gaming service providers are highly competitive because these service providers are claiming to offer high-quality graphics games with minimal internet bandwidth [40]. Popular cloud gaming service providers such as Microsoft Xbox, NVIDIA, Boosteroid, Amazon Luna, and others assert that they meet their SLAs. These companies offer their cloud gaming platforms with a minimal subscription plan, allowing users to start playing and enjoy the games after subscribing. Every gamer desires a high-quality online gaming experience with optimal performance. However, despite the good gaming content and graphics provided by many cloud games, network performance can impact the user’s QoE. Consequently, cloud service providers often collect user QoE at the end of a gaming session. The primary goal of collecting QoE data is to improve cloud gaming services. However, many players either skip filling out the QoE survey, considering it a waste of time, or fill it out without taking it seriously. Some players even pretend they are not receiving services as per the SLA, despite having a good experience, which complicates identifying the actual problems in the services.

This method employs a DL technique named EmotionNET, based on a CNN architecture. We created our own new dataset on WiFi and mobile data networks and trained it using the EmotionNET CNN model to collect emotion-based QoE. To the best of our knowledge, no similar technique exists for collecting QoE. To check our EmotionNET robustness, we trained our dataset on three different DL techniques: EfficientNET, ConvoNEXT, and ViT. After the training and testing process, we compared the efficiencies of these models’ techniques with our EmotionNET model technique. In the end, our main target was to compare the results of EmotionNET QoE on two networks (WiFi and mobile data). The main contributions of this paper are discussed below.

We created the dataset using two different network types: WiFi and 5G mobile data. First, we set up the WiFi network, selected a game from the NVIDIA cloud platform, and had our player play the game. Similarly, we selected the same game from the Boosteroid cloud platform and repeated the process.Secondly, we changed the network from WiFi to 5G mobile data and performed the same process; then we used the same game and cloud platforms.We trained the datasets from both networks on the EmotionNET model technique and subsequently compared the WiFi and 5G mobile data results.Additionally, we applied and trained three well-known models’ techniques on our custom dataset, namely ConvoNEXT, EfficientNET, and ViT. We analyzed each model’s training, validation, and testing accuracy.Likewise, we compared the performance of the three models with the EmotionNET model. We also discussed each model’s prediction accuracy using receiver operating characteristic (ROC) curves for each category and model technique.Finally, we discussed the DL-based QoE assessment on two networks (WiFi and 5G mobile data) as well as the limitations of the EmotionNET model techniques.

To overcome these challenges, we propose a novel deep learning-based QoE assessment model, EmotionNET, which evaluates user experience through facial expressions in cloud gaming. Unlike prior research, our approach does not rely on subjective surveys but instead leverages real-time emotion recognition. Furthermore, we compare EmotionNET with state-of-the-art deep learning techniques (ConvoNEXT, EfficientNET, and ViT) to validate its effectiveness.

The overall paper organization depends upon the following sections: Section 1 decribes background and related and work; Section 2 describes the methodology; Section 3 describes the results and discussions; Section 4 describes the conclusion.

## 2. Methodology

In this section, we analyzed emotion-based QoE in cloud gaming with deep learning (DL) technique, as well as applying different techniques on our generated dataset. We set up two cameras on our gaming computer: the front camera, which detects facial emotions, and the back camera, which can analyze the gaming screen. We selected 30 participants from 50 participants (age range: 18–35 years; mean age: 26.4 years) for this study. The selection criteria included active gamers with at least one year of gaming experience. Players with vision impairments or facial mobility restrictions were excluded to ensure accurate emotion recognition. First, we provided them with a hands-on testing session to familiarize them with the game. We had already notified all players about network parameters like latency, screen disturbance, and graphics imbalance. These types of condition related to player emotions were recorded. We chose Fortnite as the experimental game because it is a fast-paced multiplayer game that naturally evokes a wide range of emotions (e.g., excitement, frustration, fear, and so on). Unlike turn-based games, Fortnite requires real-time reactions and rapid decision making, which are ideal for capturing dynamic emotional responses through facial expressions. This game is very popular and does not have any horror content. During the experiment, we used two networks: WiFi and 5G mobile data. We measured both network speeds using the website (available online: https://speed.measurementlab.net/ (accessed on 24 December 2024)). The WiFi network had a download speed of 33.96 Mbps and an upload speed of 38.17 Mbps, while the mobile data network had a download speed of 21.33 Mbps and an upload speed of 12.33 Mbps. Each player played the game for 20 min under two network conditions (WiFi and 5G mobile data). To maintain consistent gameplay conditions, participants were instructed to play many rounds for 20 min of Fortnite in survival mode, engaging in standard combat interactions without intentionally altering their playstyle. Facial expressions were recorded using a Logitech C920 HD Pro Webcam, capturing video at 1080p resolution (30 FPS). The camera was positioned 50 cm away from the participant’s face, ensuring optimal lighting and visibility for emotion detection. We invited two players per day to ensure a thorough analysis of the data since each player played the same game on both cloud platforms. We have correlated the player’s emotion due to network problems via the computer screen. The deep learning models were trained for 50 epochs with a batch size of 32 and an AdamW optimizer (learning rate = 0.0001). The dataset was split into 80% training and 20% validation, and models were evaluated using cross-entropy loss. We used the EmotionNET DL technique to evaluate the facial emotions of each player. Later, we compared EmotionNET technique with three other DL techniques for ensuring that EmotionNET technique was working well on facial emotion-based dataset. Furthermore, we differentiated the player emotions related to gaming content and network disturbance with gaming screen. The overall QoE and model comparison are depicted in Figure 1.

### 2.1. Data Collection

In our data collection, we collected the videos which were formatted in mp4. This was saved automatically after finishing the game. We separated the videos which were played on WiFi network and 5G network. Furthermore, we already made the changes inside recording software to save the video after exactly 20 min. Moreover, we converted one by one videos into frames, each frame extracted at 25 frames per second (FPS). Table 2 compares the number of samples of various emotional states from two sources, NVIDIA and Boosteroid, across two types of networks: WiFi network and mobile data network. The emotional states considered are Happy, Neutral, Fearful, Disgusted, Sad, Surprised, and Angry, with all image sizes being 48 × 48 pixels. For the WiFi network, NVIDIA has the following sample counts: Happy (358,570), Neutral (205,405), Fearful (68,966), Disgusted (71,028), Sad (47,349), Surprised (95,447), and Angry (243,236). Boosteroid, on the other hand, has these sample counts: Happy (195,071), Neutral (143,691), Fearful (98,472), Disgusted (83,515), Sad (89,245), Surprised (148,921), and Angry (326,786). Regarding the mobile data network, NVIDIA has the following samples: Happy (183,853), Neutral (128,088), Fearful (96,884), Disgusted (82,177), Sad (61,708), Surprised (21,177), and Angry (489,447). Boosteroid provides these sample counts: Happy (45,922), Neutral (106,056), Fearful (96,884), Disgusted (98,911), Sad (94,780), Surprised (11,485), and Angry (556,442).

### 2.2. Preprocessing

In this section, the first recorded video was saved into the cloud server; after, our EmotionNET technique can create the image frames from the video and those image frames can be categorized as “Happy”, “Neutral”, “Fearful”, “Disgusted”, “Sad”, “Surprised”, and “Angry”. In this experiment, we used two networks, namely WiFi and mobile data internet. Furthermore, we selected two cloud platforms which are NVIDIA and Boosteroid cloud. These two clouds were accessed on WiFi and mobile data networks. Our back-end data folders have names like WiFi network (NVIDIA and Boosteroid) and mobile data network (NVIDIA and Boosteroid). Each network NVIDIA and Boosteroid folder has their seven categories; all image frames are saved into those categories. Figure 2 shows the image dataset which is categorized by emotions.

### 2.3. DL Technique Implementation

In this section, we have presented implementation details of EmotionNET technique which is based on CNN model architecture. We have made some changes into simple CNN model which is used for emotion recognition like CosineAnnealingLR Scheduler, Multiple Dropout Layers to avoid overfitting, Gradient Clipping, Custom Data Transformations, and AdamW Optimizer. The EmotionNET technique is working for facial recognition during gameplay. We have implemented this technique on custom-created facial recognition during gameplay dataset with two networks, namely WiFi and mobile data networks. This technique can categorize facial expressions into seven categorizes, namely “Angry”, “Disgusted”, “Fearful”, “Happy”, “Neutral”, “Sad”, and “Surprised”. Furthermore, we have trained and applied three more DL techniques to test our EfficientNET technique performance (ConvoNEXT, EfficientNET, and ViT) on our custom datasets.

### 2.4. Performance Evaluations

In this section, we have evaluated the four-technique performance through their training and validation accuracy, and training and validation loss. For these evaluations, we can differentiate the technique efficiencies during training and testing processes. We have explained DL technique performance evaluation details in Section 3.

### 2.5. Model Comparison

In this section, we have compared the EmotionNET technique with these three techniques (ConvoNEXT, EfficientNET, and ViT). We have collected four model techniques’ precision and recall accuracy on our custom datasets. Later, we have discussed the detail accuracy and compared with each other in Section 3.

### 2.6. QoE Comparison

In this section, we have discussed each model technique from all aspects and compared with EmotionNET model technique. Later, we have discussed the limitations of the EmotionNET model technique. Further details are explained in Section 3.

## 3. Results and Discussion

In this section, the model performance was assessed using accuracy, precision, recall, F1-score, and area under the receiver operating characteristic (ROC-AUC) curves. We also computed confusion matrices to analyze misclassifications. In addition, we performed deep analysis of each technique on our four custom datasets. Furthermore, we highlighted each technique’s precision and recall and compared with our EmotionNET technique. Moreover, we compared the analysis of each emotion technique with the EmotionNET results. At the end, we compared and discussed EmotionNET-based QoE on two networks.

### 3.1. Four Techniques’ Training and Validation Accuracy, and Training and Validation Loss

In this section, we analyzed each technique’s validation and training accuracy with their losses and overfitting. Figure 3 shows graphs for the techniques and a comparison focusing on why EmotionNET might be considered the best technique for emotion QoE analysis.

**EmotionNET:** The training accuracy steadily increases, reaching close to 0.99 by the end of 50 epochs, while the validation accuracy also improves consistently, stabilizing around 0.88. This indicates that the model is learning well from the training data and generalizing effectively on unseen data. The training loss decreases sharply, settling around 0.26, while the validation loss decreases as well but at a slower rate, reaching about 0.34. The gap between the training and validation loss is relatively small, suggesting that the model is not significantly overfitting.

**ConvoNEXT:** The training accuracy quickly reaches 0.99 within the first few epochs. However, the validation accuracy stands around 0.92, indicating a strong performance but possibly a minor generalization gap. The training loss drops significantly, nearing 0, which is typical for a model that fits very well to the training data. However, in the beginning, validation loss is decreasing but, after 10 epochs, it starts to increase and shows that the model is overfitting.

**EfficientNET:** The training accuracy also reaches well like ConvoNEXT and the validation accuracy reaches 0.92. In this model, it is the same problem of overfitting because in the beginning it is decreasing and learning well but after 10 epochs it starts increasing. This is a clear indication that the model is overfitting and no longer improving in general terms.

**ViT:** In this technique, the training accuracy improves rapidly and achieves almost perfect accuracy. The validation accuracy of this ViT technique reaches 0.91; this shows that model works and learns well. However, we looked into the training loss which is decreasing quickly like the other techniques. Moreover, in the beginning, the validation loss decreases but when it reaches 10 epochs it starts increasing which means the model technique is overfitting.

**Comparative analysis of training and validation accuracy with their loss:** We tested four model techniques on our custom-created dataset. The training loss and validation loss show that the ConvoNEXT, EfficientNET, and ViT model techniques are overfitting in our custom emotion recognition dataset. EmotionNET performs better and balanced accuracy between the training and validation. Although the validation and training accuracy is not much better than ConvoNEXT and EfficientNET, it has the smallest gap between the training and validation loss. This means that the EmotionNET technique performs better on emotion recognition datasets and it has less overfitting as compared to the other model techniques. The other model techniques have high training accuracy but they suffer from increased validation loss which means the model techniques are overfitting. They have limitations on their effectiveness and unseen data. From the graph and data, the EmotionNET model technique shows the best and most balanced performance among the four model techniques because it became balanced between the learning and validation of emotion recognition data.

### 3.2. Four Model Techniques’ Performance Analysis and Comparison

In this section, we evaluate the performance of the four model techniques in binary classification. The ROC curves help us to understand each technique performance in depth with their positive and negative classes.

**EmotionNET Model Technique ROC:** The ROC curves in Figure 4 show the classification performance of the EmotionNET model technique across six emotion categories (Angry, Disgusted, Fearful, Happy, Neutral, Sad, and Surprised) for four datasets: the WiFi (Boosteroid and NVIDIA) datasets, and the mobile data (Boosteroid and NVIDIA) datasets. Each emotion category shows varying levels of classification accuracy, measured by the AUC (area under the curve), where a higher AUC value indicates better performance. For the “Angry” emotion, the EmotionNET technique performs excellently across three datasets, with AUC values close to 0.99, but the mobile data NVIDIA dataset shows much lower performance, with an AUC of 0.78.

Similarly, for the “Disgusted” emotion, this technique achieves AUC values above 0.96 for most datasets, while the mobile data NVIDIA dataset lags with an AUC of 0.84. In the case of the “Fearful” emotion, three datasets again perform strongly, achieving near-perfect AUC values around 0.99 to 1.00, while the mobile data NVIDIA dataset records a significantly lower AUC of 0.70. The pattern is similar for the “Happy” emotion, where AUC values for three datasets are as high as 0.99 to 1.00, while the mobile data NVIDIA dataset lags behind at 0.76. The “Neutral” emotion shows similarly high AUC values across most datasets, ranging from 0.98 to 0.99, except for the mobile data NVIDIA dataset, which achieves a moderate AUC of 0.81. For the “Sad” emotion, the model performs well on the WiFi (Boosteroid and NVIDIA), and mobile data Boosteroid datasets (AUC = 0.98), but the mobile data NVIDIA dataset records its lowest performance overall, with an AUC of 0.65. Finally, the “Surprised” emotion follows the general trend, with high AUC values around 0.99 for most datasets, while the mobile data NVIDIA dataset performs comparatively lower with an AUC of 0.85. The EmotionNET model technique demonstrates a strong classification performance for most datasets, with AUC values close to 1.0 for many emotions. However, the mobile data NVIDIA dataset consistently underperforms across all emotion categories, suggesting it may have more challenging data or lower quality, making it harder for the model to classify emotions accurately. The EmotionNET model demonstrates superior performance in a low-latency (WiFi) environment, achieving high precision and recall across most emotional categories. However, under high-latency conditions (mobile data, especially the NVIDIA dataset), the classification performance dropped significantly, particularly for emotions such as “Disgusted” and “Surprised”. The increased latency likely caused a delay in facial expression responses, leading to misclassifications. While the model performed better than other deep learning techniques in real-time gaming scenarios, future enhancements are required to adapt to dynamic network conditions.

**ConvoNEXT Model Technique ROC:** Figure 5 highlights the ROC curves for different emotion categories across four datasets. Each ROC curve represents the performance of a classification model for a specific emotion category: Angry, Disgusted, Fearful, Happy, Neutral, Sad, and Surprised. The datasets include the following: WiFi Boosteroid consistently demonstrates excellent performance across all emotion categories. For Angry, Fearful, Sad, and Surprised, the AUC values are approximately 0.98 to 0.99, indicating that the model can effectively classify these emotions with minimal false positives. The model performs exceptionally well for Neutral and Happy, achieving an AUC of 1.00, which suggests near-perfect classification for these emotions. Even for the more challenging category of Disgusted, the model maintains a high AUC of 0.96, further emphasizing the reliability of this dataset in producing accurate emotion detection results. Similarly to the WiFi Boosteroid dataset, the mobile data Boosteroid dataset shows a strong classification performance across all emotions. For emotions such as Angry, Happy, Sad, and Neutral, the AUC values range from 0.97 to 0.99, which indicates excellent model performance. The model handles challenging emotions like Fearful and Disgusted well, with AUC values around 0.98. While slightly lower for Surprised, with an AUC of 0.90, the dataset still performs effectively, showing the model’s ability to generalize well across different emotion categories.

The WiFi NVIDIA dataset exhibits a significant drop in performance compared to the Boosteroid datasets. For emotions such as Angry, Fearful, and Sad, the AUC values fall between 0.67 and 0.83, reflecting moderate-to-poor classification accuracy. The ROC curves for these emotions are much closer to the diagonal line, suggesting that the model struggles to distinguish between different emotional states in this dataset. In particular, for the Disgusted category, the performance is notably weak, with the model unable to effectively differentiate this emotion from others. While it performs slightly better for Happy and Neutral with AUC values close to 0.83, the overall performance remains suboptimal across all emotion categories. The mobile data NVIDIA dataset, like its Boosteroid equal, shows a high classification performance for most emotions. Emotions like Angry, Fearful, Sad, Happy, and Neutral have AUC values between 0.97 and 0.99, signifying excellent classification accuracy with few false positives. However, the performance for Surprised is slightly lower, with an AUC of 0.90, indicating a small drop in accuracy. The model performs similarly well for Disgusted, maintaining an AUC of around 0.98, suggesting the model can handle complex emotions in this dataset effectively. The WiFi Boosteroid dataset outperforms the others, especially for emotions like Neutral and Happy, where it achieves an AUC of 1.00. The mobile data (Boosteroid and NVIDIA) datasets also show a strong performance across most emotions. In contrast, the WiFi NVIDIA dataset exhibits a significantly weaker performance, particularly for emotions like Angry, Fearful, and Disgusted, where the model struggles to classify emotions accurately.

**EfficientNET Model Technique ROC:** Similarly, Figure 6 contains ROC curves for seven emotional categories: Angry, Disgusted, Fearful, Happy, Sad, Neutral, and Surprised. The performance in the WiFi Boosteroid dataset is moderate. For emotions such as Angry and Fearful, the AUC values are 0.78 and 0.76, indicating a reasonable classification accuracy but with room for improvement. For Happy and Neutral, the AUC scores are 0.96 and 0.81, representing a much better performance, especially for the Happy emotion, where the model exhibits excellent classification capability. However, for Disgusted, the model’s performance is weak with an AUC of 0.43, showing considerable difficulties in distinguishing this emotion. The Sad and Surprised emotions show a reasonable performance, with AUC values of 0.64 and 0.70, respectively, highlighting that the model handles these categories moderately well but could improve. The WiFi NVIDIA dataset shows a notable drop in performance compared to the others. For the Angry and Sad emotions, the AUC values are 0.74 and 0.64, showing a moderate classification accuracy but a significant distance from ideal. For Fearful, the AUC is 0.84, indicating the model performs relatively better, with fewer false positives and improved sensitivity. The Happy emotion is one of the strongest in this dataset, with an AUC of 0.94, but Disgusted remains poorly classified, with a low AUC of 0.48, showing considerable difficulty in distinguishing this emotion from others. For Neutral and Surprised, the AUC values are 0.64 and 0.65, respectively, suggesting the model has trouble accurately predicting these emotions in this dataset.

The mobile data Boosteroid dataset performs quite well in this dataset for most emotions. For Angry, the AUC is 0.67, reflecting some challenges in classification, but it improves significantly for Fearful with an AUC of 0.81. The Happy and Neutral emotions exhibit a strong performance with AUC values of 0.97 and 0.81, indicating that the model can classify these emotions accurately, with few false positives. For Disgusted, the model continues to struggle, achieving an AUC of 0.49, similar to other datasets. The AUC for Sad is 0.70, showing moderate classification, while Surprised achieves a score of 0.64, indicating room for improvement. The mobile data NVIDIA dataset shows the highest performance for many emotions. For Angry, the AUC is 0.90, indicating a strong classification accuracy with minimal false positives. For Fearful, the model also performs well, with an AUC of 0.90, and, for Happy, it achieves an AUC of 0.98, showing almost perfect classification for this emotion. The model has a better handle on Disgusted in this dataset compared to others, with an AUC of 0.81, which is a noticeable improvement. For Neutral and Sad, the AUC values are 0.90 and 0.97, respectively, indicating a high performance, particularly for Sad. Surprised also achieves a relatively good score with an AUC of 0.90, showing that the model performs strongly across most emotions in this dataset.

**ViT Model Technique ROC:** Furthermore, Figure 7 represented the ROC curves for seven dataset emotion categories (“Angry”, “Disgusted”, “Fearful”, “Happy”, “Neutral”, “Sad”, and “Surprised”) across four different datasets. The ROC curve is a graphical representation of a model’s ability to distinguish between classes, with the True Positive Rate plotted against the False Positive Rate. The AUC is also provided as a metric to summarize the overall performance of each model, with values closer to 1.0 indicating better performance. The WiFi Boosteroid dataset demonstrates a consistently strong performance across all emotions. For Angry, Fearful, Sad, Happy, Neutral, and Surprised, the AUC values range from 0.96 to 0.99, indicating a very high classification accuracy with minimal false positives. For Disgusted, the AUC is slightly lower at 0.97, but still suggests an excellent performance, especially compared to the other datasets. The ROC curve is close to the ideal top-left corner for all emotions, suggesting a reliable model in this dataset. The WiFi NVIDIA dataset shows a more varied performance. For Angry, the AUC is 0.94, indicating a good but slightly lower classification accuracy than the WiFi Boosteroid dataset. For Fearful, Neutral, and Happy, the AUC values range between 0.95 and 0.98, reflecting an excellent classification performance. For Disgusted, the model struggles, with an AUC of 0.60, highlighting the difficulty in distinguishing this emotion in this dataset. The ROC curve for Disgusted is far from the top-left corner, indicating a higher rate of false positives. The Sad and Surprised emotions have strong AUC values of 0.96 and 0.94, showing high reliability for these categories.

The mobile data Boosteroid dataset displays a solid performance across all emotions. For Angry, Fearful, Sad, Neutral, and Happy, the AUC values range between 0.97 and 1.00, demonstrating near-perfect classification, especially for Happy, where the AUC reaches 1.00. The performance for Disgusted is strong, with an AUC of 0.98, showcasing the model’s ability to handle this challenging emotion well in this dataset. Surprised has a slightly lower AUC of 0.97, but still indicates highly accurate classification with minimal false positives. The mobile data NVIDIA dataset shows mixed results, with a strong classification for Angry (AUC 0.85) and Happy (AUC 0.90). The model performs moderately for Fearful and Neutral, with AUC values of 0.85 and 0.81, respectively, indicating reasonable classification but with some room for improvement. For Disgusted, the model struggles significantly, achieving the lowest AUC at 0.40, suggesting a high rate of misclassification. For Sad and Surprised, the AUC values are 0.76 and 0.86, respectively, showing a moderate performance but significantly lower than the other datasets.

**ROC-based model technique comparison:** EmotionNET stands out as the top- performing model across all emotions and datasets, demonstrating a consistently high performance, particularly in the “Happy”, “Sad”, and “Neutral” categories, where it achieves near-perfect AUC scores. Although it excels overall, EmotionNET, like the other models, encounters challenges in correctly classifying the “Disgusted” emotion, indicating that this emotion is inherently more difficult to predict across different datasets. In comparison, ConvoNEXT also performs well but exhibits more variation in AUC scores across different emotions. It competes closely with EmotionNET in categories like “Happy” and “Fearful” but generally falls short in others, such as “Angry” and “Neutral”. EfficientNET shows significant variability across emotions, with its performance being less consistent than both EmotionNET and ConvoNEXT. While it performs adequately in “Angry” and “Sad”, it struggles notably with the “Disgusted” emotion. Lastly, ViT shows a strong performance in several emotions, particularly in “Happy” and “Sad”, sometimes matching or even surpassing EmotionNET. However, ViT also exhibits more pronounced fluctuations in performance across different emotions, similar to ConvoNEXT and EfficientNET. EmotionNET is the best choice for deployment, given its superior and more consistent performance across all emotions, though all models show a common weakness in classifying the “Disgusted” emotion, highlighting an area for potential improvement. EmotionNET demonstrated superior accuracy and generalizability with minimal overfitting. ConvoNEXT and EfficientNET, while achieving higher training accuracy, suffered from overfitting, making them less effective in real-world scenarios. ViT exhibited an inconsistent classification performance across different network conditions, suggesting that transformer-based architectures may require additional fine-tuning for emotion recognition in QoE assessment.

### 3.3. Precision and Recall of Four Model Techniques on Four Custom Datasets

In this section, we have analyzed the four models-technique’s precision and recall performance. It shows the each technique’s performance on our custom dataset.

**EmotionNET model technique prediction performance:** In the Table 3 WiFi Boosteroid dataset, the model technique demonstrates high precision and recall across most emotion categories, with “Angry” having a precision of 0.996451 and a recall of 0.943507. The “Disgusted” category shows a precision of 0.998307 and a recall of 0.894303, suggesting strong but slightly lower recall. For “Fearful”, the model achieves a precision of 0.8966 and a recall of 0.958045, indicating high recall but some false positives. “Happy” is detected with a precision of 0.980498 and a recall of 0.946979, showing a strong overall performance. The “Neutral” category, while having a high recall of 0.992221, shows a lower precision of 0.792283, possibly due to misclassification with other emotions. “Sad” is well balanced with a precision of 0.948759 and a recall of 0.933745. However, the “Surprised” category has a lower precision of 0.509869 but a high recall of 0.932985, indicating frequent detection but some confusion with other emotions. Overall, the model achieves an accuracy of 0.94432, with a macro average precision of 0.874681 and recall of 0.943112, and a weighted average precision of 0.954899 and recall of 0.94432. For the WiFi NVIDIA dataset, the model maintains an excellent performance, especially in the “Angry” category, with a precision of 0.992839 and a recall of 0.940402. The “Disgusted” category shows near-perfect precision at 0.999209, though with a slightly lower recall of 0.845461. “Fearful” is detected with a precision of 0.903822 and a recall of 0.95817, reflecting good detection with some false positives. The “Happy” category exhibits a strong performance with a precision of 0.996376 and a recall of 0.941228. The “Neutral” category has a slightly lower precision of 0.837867 but a high recall of 0.99181. The “Sad” category shows balanced detection with a precision of 0.79293 and a recall of 0.932435. However, “Surprised” has lower precision at 0.684749 but high recall at 0.947592. The accuracy for this dataset is 0.94901, with a macro average precision of 0.886828 and recall of 0.936854, and a weighted average precision of 0.949317 and recall of 0.94901.

In the mobile data Boosteroid dataset, this technique performs well, particularly in the “Angry” category, with a precision of 0.997275 and a recall of 0.912834. The “Disgusted” category also shows high precision at 0.997842 but slightly lower recall at 0.8358. For “Fearful”, the model achieves a precision of 0.939096 and a recall of 0.952018, indicating strong detection. “Happy” is detected with near-perfect precision of 0.993108 and a recall of 0.97137. The “Neutral” category, while having a very high recall of 0.992797, shows a lower precision of 0.797852. The “Sad” category exhibits good detection with a precision of 0.837156 and a recall of 0.967115. “Surprised” is detected with both high precision of 0.93585 and recall of 0.927648. Overall, the model achieves an accuracy of 0.937588, with a macro average precision of 0.930802 and recall of 0.937088, and a weighted average precision of 0.945943 and recall of 0.937588. However, in the mobile data NVIDIA dataset, the model’s performance significantly declines. The “Angry” category shows a precision of 0.585788 and a recall of 0.234193, indicating poor performance. The “Disgusted” category has both precision and recall at 0, suggesting a complete failure in detection. “Fearful” is detected with a precision of 0.237341 and a recall of 0.417283, showing low performance. The “Happy” category has better precision at 0.893458, though recall is still low at 0.490776. The “Neutral” category shows good precision at 0.949068 but lower recall at 0.83284. The “Sad” category is detected with a precision of 0.120353 and a recall of 0.25301, indicating low performance. The “Surprised” category also performs poorly, with a precision of 0.490042 and a recall of 0.044269. The overall accuracy for this dataset is 0.407133, with a macro average precision of 0.374598 and recall of 0.320782, and a weighted average precision of 0.543385 and recall of 0.407133, reflecting weak detection capability across categories. Given in Figure 8 is the graphical representation of the precision and recall performance in the EmotionNET model. The classification results show that EmotionNET’s performance varies based on network stability. The model maintained high AUC scores (0.99) on WiFi networks, indicating accurate emotion recognition. However, in mobile data conditions, the AUC scores dropped (e.g., 0.70 for “Fearful” and 0.76 for “Happy” in the NVIDIA dataset), confirming that unstable latency impacts facial emotion-based QoE estimation. The results suggest that latency-aware adaptation mechanisms should be integrated into future model versions.

**ConvoNEXT model technique prediction performance:** In Table 4, the ConvoNEXT model technique’s performance varies across different datasets, as demonstrated by the evaluation metrics provided. In the WiFi Boosteroid dataset, the model shows high precision for the “Angry” category at 0.975075, with a recall of 0.76526, indicating good accuracy in predicting “Angry” emotions but with some false negatives. The “Disgusted” category achieves both extremely high precision and recall, at 0.999612 and 0.930386, respectively, reflecting the model’s excellent performance in this category. For “Fearful” emotions, the precision is lower at 0.652562, but recall is higher at 0.807772, meaning the model detects most instances of this emotion, albeit with some classification errors. The “Happy” category also shows a strong performance with a precision of 0.892505 and a recall of 0.827109. However, the model’s performance in the “Neutral” category is mixed, with a low precision of 0.405371 but an exceptionally high recall of 0.977298, suggesting that while many instances are classified as “Neutral”, not all are correct. The “Sad” category has moderate precision and recall values of 0.672179 and 0.733344, respectively. The model struggles most with the “Surprised” category, where precision and recall are the lowest at 0.443454 and 0.569191, respectively. Overall, the model’s accuracy for this dataset is 0.752011, with a macro average precision of 0.72012 and recall of 0.723694, and a weighted average precision of 0.848551. In the WiFi NVIDIA dataset, the model shows slightly higher precision for the “Angry” category at 0.931996 but a lower recall of 0.7507. For the “Disgusted” category, while precision remains high at 0.99824, recall significantly drops to 0.241543, indicating that many instances go undetected. The “Fearful” category sees an improvement in precision to 0.678374, with a substantial increase in recall to 0.800021. The “Happy” category maintains very high precision at 0.979006 and recall at 0.819233, demonstrating a strong performance. In the “Neutral” category, precision drops to 0.467218, though recall remains high at 0.972754. The “Sad” category shows a moderate performance with a precision of 0.502351 and a recall of 0.761708, while the “Surprised” category has a precision of 0.618405 and a recall of 0.549401. The accuracy for this dataset is 0.75211, with a macro average precision of 0.73937 and recall of 0.705863, and a weighted average precision of 0.834592.

In the mobile data Boosteroid dataset, this technique maintains high precision for the “Angry” category at 0.981519, but recall drops to 0.464809. For the “Disgusted” category, both precision and recall remain high at 0.997433 and 0.26333, respectively, indicating accurate but less sensitive performance. The “Fearful” category has moderate precision at 0.696016 and higher recall at 0.756588. Precision and recall are high in the “Happy” category, at 0.985466 and 0.807506, respectively. The “Neutral” category shows low precision at 0.321687 but high recall at 0.973237, suggesting the model is more sensitive but less accurate in this category. The “Sad” category shows a balanced performance, with precision and recall at 0.540343 and 0.863147, respectively. However, the model struggles with the “Surprised” category, where both precision and recall are low, at 0.349073 and 0.341246, respectively. The overall accuracy for this dataset drops to 0.622849, with a macro average precision of 0.775363 and recall of 0.641776, and a weighted average precision of 0.826255. Finally, in the mobile data NVIDIA dataset, the model’s performance declines further, with precision dropping to 0.645119 and recall to 0.221551 for the “Angry” category. The “Disgusted” category maintains high precision at 0.997433 but very low recall at 0.26333. Precision in the “Fearful” category drops significantly to 0.214528, with a recall of 0.484776. The “Happy” category has high precision at 0.97986 but lower recall at 0.508122. For the “Neutral” category, precision increases to 0.913174, and recall is very high at 0.879558. Both precision and recall are low in the “Sad” category, at 0.214528 and 0.246687, respectively. The “Surprised” category also shows low precision and recall, at 0.154091 and 0.119015, respectively. The overall accuracy for this dataset further decreases to 0.609476, with a macro average precision of 0.436237 and recall of 0.374643, and a weighted average precision of 0.609476. This indicates a significantly lower performance on the mobile data NVIDIA dataset, particularly for certain emotion categories. Furthermore, Figure 9 shows the graphical representation of the ConvoNEXT model technique’s precision and recall.

**EfficientNET model technique prediction performance:** Table 5 presents the performance of an EfficientNET model technique evaluated across four datasets: WiFi (Boosteroid and NVIDIA) and mobile data (Boosteroid and NVIDIA). For the WiFi Boosteroid dataset, the model shows high precision in the “Angry” category (0.900671) but low recall (0.319717). The “Fearful” category has a lower precision (0.231446) but a relatively high recall (0.582723), indicating better detection ability but lower precision. The “Happy” category shows a balanced performance with precision and recall values of 0.691593 and 0.637561, respectively. However, categories like “Disgusted” and “Surprised” perform poorly, with both precision and recall at 0 for “Disgusted” and low values for “Surprised” (precision: 0.081096, recall: 0.121697). The accuracy for this dataset is 0.373277, with a macro average precision of 0.34118 and recall of 0.392742. The weighted average precision is 0.598176, with a recall of 0.373277. For the WiFi NVIDIA dataset, the “Angry” category achieves a precision of 0.864387 and a recall of 0.293137. The “Fearful” category shows a precision of 0.253531 and a recall of 0.573052, while the “Happy” category performs well with a precision of 0.907626 and a recall of 0.63115. The “Neutral” category also shows high recall (0.850652) but lower precision (0.241137). Again, “Disgusted” has no detection, and “Surprised” has limited detection ability (precision: 0.114437, recall: 0.117328). The accuracy for this dataset is 0.411646, with a macro average precision of 0.362561 and recall of 0.37906, and a weighted average precision of 0.617448 with a recall of 0.411646.

In the mobile data Boosteroid dataset, the “Angry” category has a precision of 0.828677 but a very low recall of 0.054913. The “Fearful” category shows moderate performance with precision and recall values of 0.360429 and 0.546636, respectively. The “Happy” category performs well with a precision of 0.930001 and recall of 0.670511. The “Neutral” category has a high recall (0.909071) but low precision (0.19181). The “Surprised” category performs poorly with a precision of 0.502004 and a recall of 0.038783. The overall accuracy for this dataset is the lowest among the four at 0.339924, with a macro average precision of 0.452749 and recall of 0.368804. The weighted average precision is 0.570115 with a recall of 0.339924. Lastly, the mobile data NVIDIA dataset demonstrates the best overall performance. The “Angry” category achieves a precision of 0.983136 and a recall of 0.715405. The “Fearful” category shows high precision (0.65981) and recall (0.955705), while the “Happy” category has a very strong performance with precision and recall values of 0.983295 and 0.944172, respectively. The “Neutral” category also performs well with a precision of 0.727481 and a recall of 0.96838. The “Surprised” category has a precision of 0.830032 and a recall of 0.775642. The overall accuracy for this dataset is the highest at 0.867067, with a macro average precision of 0.849962 and recall of 0.818839, and a weighted average precision of 0.899558 with a recall of 0.867067. Moreover, Figure 10 shows the graphical representation of the EfficientNET model technique precision and recall performance.

**ViT model technique prediction performance:** In Table 6, for the WiFi Boosteroid dataset, the model technique performed exceptionally well with high precision and recall across most categories. Specifically, this technique achieved a precision of 0.9957 and recall of 0.9133 for the “Angry” category, 0.9999 precision and 0.8943 recall for “Disgusted”, and 0.9859 precision with 0.9551 recall for “Happy”. The “Neutral” category had a precision of 0.7187 but a high recall of 0.9888. The overall accuracy was strong at 0.9249, with macro averages of 0.8611 in precision and 0.9340 in recall, and weighted averages of 0.9411 in precision and 0.9249 in recall. The WiFi NVIDIA dataset also demonstrated good performance, with “Happy” showing 0.9659 precision and 0.9524 recall. The “Angry” category had a precision of 0.9722 and recall of 0.9089. However, the performance dropped for “Fearful” with a precision of 0.8554 and recall of 0.5058, and for “Disgusted”, where precision was 0.9993, but recall was only 0.6832. The model achieved an accuracy of 0.9141, with macro averages of 0.8756 in precision and 0.9065 in recall, and weighted averages of 0.9277 in precision and 0.9141 in recall.

In the mobile data Boosteroid dataset, the ViT model technique maintained a high performance, particularly in the “Happy” category with a precision of 0.9975 and recall of 0.9190. The “Angry” category also showed strong results with a precision of 0.9959 and recall of 0.8688. The overall accuracy was 0.9167, with macro averages of 0.9152 in precision and 0.9223 in recall, and weighted averages of 0.9340 in precision and 0.9167 in recall. However, in the mobile data NVIDIA dataset, the model’s performance declined significantly. The “Angry” category had a precision of 0.6257 and recall of 0.1249, while the “Disgusted” category failed entirely, registering 0 for both precision and recall. The “Fearful” category also showed low performance, with a precision of 0.2118 and recall of 0.4059. Despite this, the “Happy” category remained strong with a precision of 0.9757 and recall of 0.9587. The dataset’s overall accuracy was the lowest at 0.4731, with macro averages of 0.4197 in precision and 0.3686 in recall, and weighted averages of 0.5979 in precision and 0.4731 in recall. At the end, Figure 11 shows the graphical representation of the ViT model precision and recall.

**Four Model techniques’ Comparison with EmotionNET prediction performance:** Upon comparing the precision and recall performance across the EmotionNET, ConvoNEXT, EfficientNET, and ViT model techniques, it is evident that EmotionNET consistently outperforms the others. EmotionNET demonstrates exceptionally high precision across a broad range of emotions, particularly excelling in categories like “Happy”, “Neutral”, and “Surprised”. where it nearly achieves perfect scores. Even in more challenging categories, such as “Disgusted”, where other models tend to struggle, EmotionNET still maintains competitive precision, highlighting its robustness. Conversely, ConvoNEXT, while performing well overall, shows a noticeable dip in precision for emotions like “Disgusted”, “Sad”, and “Surprised”, making it slightly less reliable than EmotionNET. Similarly, EfficientNET and ViT exhibit solid precision for some emotions, but they also suffer from significant drops, particularly for “Disgusted” and “Fearful”, where their precision is markedly lower than that of EmotionNET.

In terms of recall, EmotionNET continues to lead, displaying high recall across all emotions, which underscores its effectiveness in accurately identifying and capturing the relevant instances of each emotion. Although there is a minor drop in recall for the “Disgusted” emotion, EmotionNET still outperforms the other models, which exhibit more pronounced declines in recall for this and other emotions. ConvoNEXT, while balanced in its recall performance, struggles with emotions like “Disgusted” and “Fearful”, mirroring its precision shortcomings. EfficientNET also falters in recall for these emotions, particularly “Sad”, where it falls behind EmotionNET. ViT, though competitive in certain aspects, similarly suffers from inconsistent recall, particularly for “Disgusted”, “Sad”, “Fearful”, and “Surprised”. EmotionNET emerges as the best-performing model in this comparison. Its consistent and high performance in both precision and recall across various emotions makes it the most reliable model technique among those evaluated, especially in handling more challenging emotional categories where others fall short. This demonstrates the robustness and superior accuracy of EmotionNET in emotion detection tasks. We observed a significant drop in model performance under mobile data conditions, particularly for the NVIDIA dataset. The results indicate that real-time latency fluctuations impact facial expression recognition, leading to lower QoE prediction accuracy. This suggests that deep learning models should be optimized for network variability to maintain robust emotion-based QoE assessment.

### 3.4. Analysis of Emotions and QoE Comparison

During the experiment, we analyzed the player’s emotions while playing the online game from those clouds. After the assessment of players’ emotions, we realized that network performance was affecting players’ QoE, because the same players played the “Fortnite” game on both networks and we collected different emotions while playing the game.

Figure 12 provides a comprehensive comparison of detected emotions across four different datasets: WiFi (Boosteroid and NVIDIA) and mobile data (Boosteroid and NVIDIA). In the mobile data Boosteroid dataset, the “Angry” emotion is the most dominant, with a detection rate of 52.1%, indicating a significant presence of negative emotions. The “Disgusted” emotion is moderately detected at 8.9%, while “Fearful” is also prominent at 10.2%. Positive emotions like “Happy” are under-represented, with only 4.4% detection, making it the least detected emotion in this dataset. The “Neutral” emotion is detected at a moderate rate of 13.1%, reflecting a balanced emotional state, while “Sad” shows a slightly higher detection at 9.2%. The “Surprised” emotion is minimally detected at 2.1%, indicating it plays a minor role in this dataset.

In the mobile data NVIDIA dataset, the “Angry” emotion remains highly detected at 43.5%, though slightly less than in the WiFi Boosteroid dataset. The “Disgusted” emotion is consistent with a detection rate of 8.8%, while “Fearful” is slightly less prominent at 9.9%. Notably, the “Happy” emotion shows a significant increase to 16.3%, indicating better representation of positive emotions. The “Neutral” emotion is moderately present at 14.2%, similar to the first dataset. The “Sad” emotion shows a slightly higher detection at 9.5%, suggesting more varied emotions in this dataset. The “Surprised” emotion is detected slightly more than in the first dataset at 2.7%, though it still remains under-represented.

The WiFi Boosteroid dataset shows a significant drop in the detection of “Angry” emotions, with a rate of 27.4%, indicating a shift in emotion dominance. The “Disgusted” emotion is detected less frequently at 6.8%, showing a reduction in negative emotions. Interestingly, the “Fearful” emotion shows a slight increase to 11.1%, indicating a stronger presence of fear-related emotions. The “Happy” emotion continues to increase, reaching 17.5%, making it one of the more dominant emotions in this dataset. The “Neutral” emotion shows a slight increase to 16.4%, suggesting a balanced emotional state. The “Sad” emotion is consistent at 9.1%, while the “Surprised” emotion shows a significant increase to 13.3%, making it a notable emotion in this dataset.

The WiFi NVIDIA dataset presents a dramatic shift in emotion detection, with the “Angry” emotion almost disappearing at a mere 0.3%. The “Disgusted” emotion is slightly less prominent at 6.5%, maintaining a minor role. The “Fearful” emotion is consistently present at 9.2%, though not dominant. The “Happy” emotion reaches its highest detection rate in this dataset at 18.1%, marking a shift towards more positive emotions. The “Neutral” emotion becomes overwhelmingly dominant, with a detection rate of 51.7%, indicating a strong neutral state in this dataset. The “Sad” emotion remains consistent with a detection rate of 9.1%, similar to other datasets. However, the “Surprised” emotion experiences a sharp drop to 0.7%, making it almost not important in this dataset.

When we applied the DL-based technique on emotion recognition, it gave remarkable results in emotion-based QoE. Later, we compared the EmotionNET technique with other DL-based techniques on our custom-created datasets. During the training process, we found that other models like ConvoNEXT, EfficientNET, and ViT have quite good training accuracy but they have a big overfitting problem. We found that the EmotionNET technique is the best technique to analyze emotion-based QoE. After the detailed observation of the EmotionNET technique on a custom emotion-based dataset, it is clearly visible that the network is affecting the player’s QoE, because the WiFi network has stable connectivity as compared to mobile data. This type of QoE helps cloud service providers to know the accurate QoE. These findings have significant implications for cloud gaming services. Emotion-based QoE assessment provides a more objective measure of user experience compared to traditional surveys. Cloud gaming providers can use this technology to monitor real-time user satisfaction and dynamically adjust video quality, latency compensation, or server allocation to optimize the gaming experience. Moreover, integrating this approach into cloud gaming platforms can lead to enhanced user engagement and reduced churn rates.

## 4. Conclusions

The main objective of this study was to assess the QoE in cloud gaming of a VR environment using deep learning models. During this experiment, players’ facial expressions were detected while playing the game. Our results show that the EmotionNET model works better than the other three models. The EmotionNET model predicts human emotions very efficiently and it has minor overfitting as compared to the other models. Furthermore, facial expression-based QoE assessment using the EmotionNET model can be a beneficial solution for cloud service providers because they will easily identify more accurately players’ QoE. Traditional methods always give us reduced and inaccurate information regarding user satisfaction level towards online gaming. This approach helps to improve the actual gaming service quality and customer satisfaction. Despite the favorable results, this study has some limitations. First of all, this study needs to use a more than 5 million image dataset because we trained and tested this model on 30 players with 20 min of each game session. Second, this model technique may not give accurate QoE for a group of people because a group of people will give different facial expressions; these limitations can be overcome in future work. This study confirms that EmotionNET effectively predicts facial emotion-based QoE in cloud gaming, especially in low-latency environments (WiFi). However, the model’s performance degrades under high-latency conditions, particularly in the mobile data NVIDIA dataset. Although the EmotionNET model demonstrates a strong performance in low-latency cloud gaming environments, it exhibits reduced accuracy under high-latency network conditions, particularly in the mobile data NVIDIA dataset. The model faces challenges in classifying emotions such as “Disgusted” and “Surprised” due to network-induced expression delays. Future work will focus on developing latency-aware adaptation techniques to improve real-time robustness. Additionally, integrating physiological signals (e.g., heart rate, gaze tracking) with facial expressions may enhance the reliability of QoE assessments under dynamic cloud gaming conditions. Furthermore, deploying EmotionNET as an edge AI model could reduce network transmission delays, making it more suitable for real-time gaming applications. In the end, this type of study offers a novel and effective method of QoE assessment in cloud gaming. Because of the high demand of cloud gaming, it is important to collect high levels of QoE assessment and user satisfaction.

## Figures and Tables

**Figure 1 sensors-25-01594-f001:**
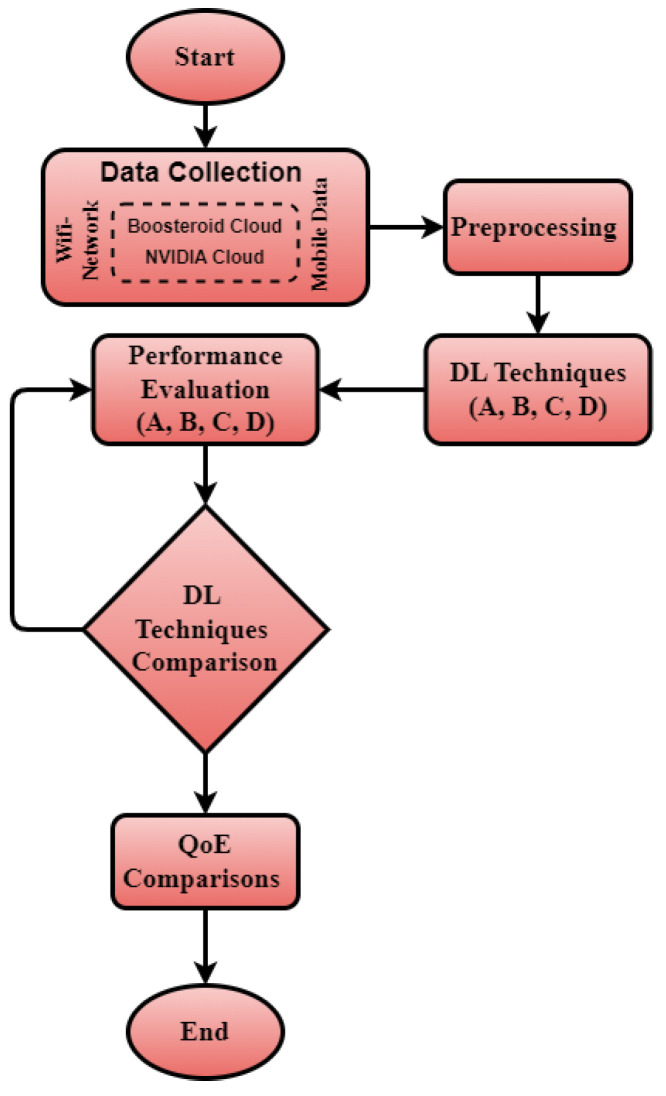
Structure of our QoE comparison.

**Figure 2 sensors-25-01594-f002:**
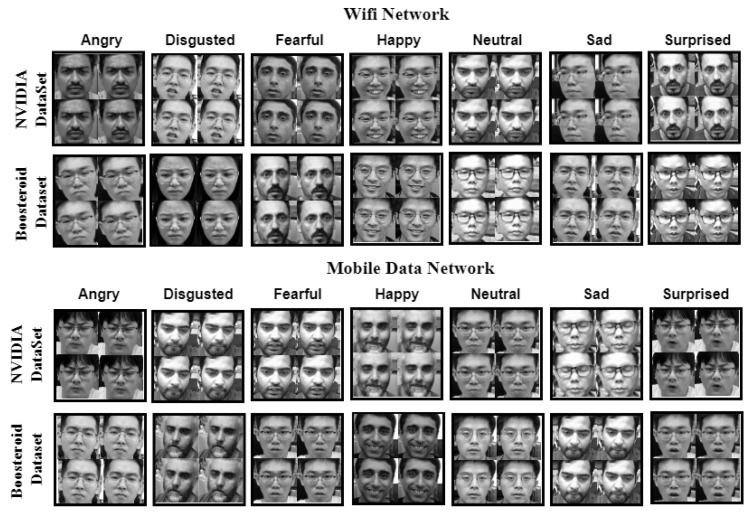
Image dataset collected on two networks.

**Figure 3 sensors-25-01594-f003:**
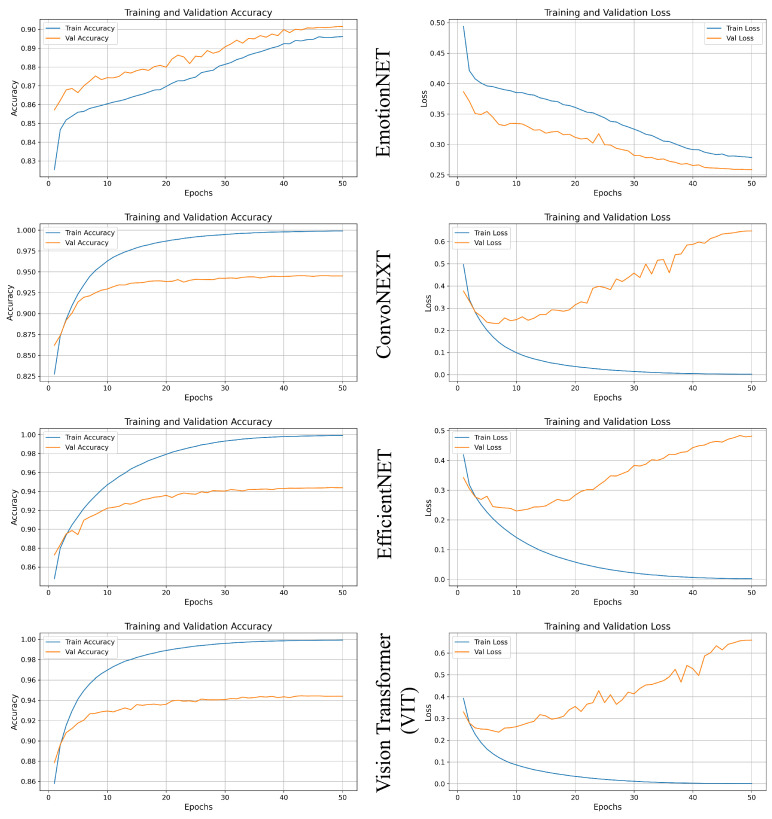
Four models’ accuracy and loss.

**Figure 4 sensors-25-01594-f004:**
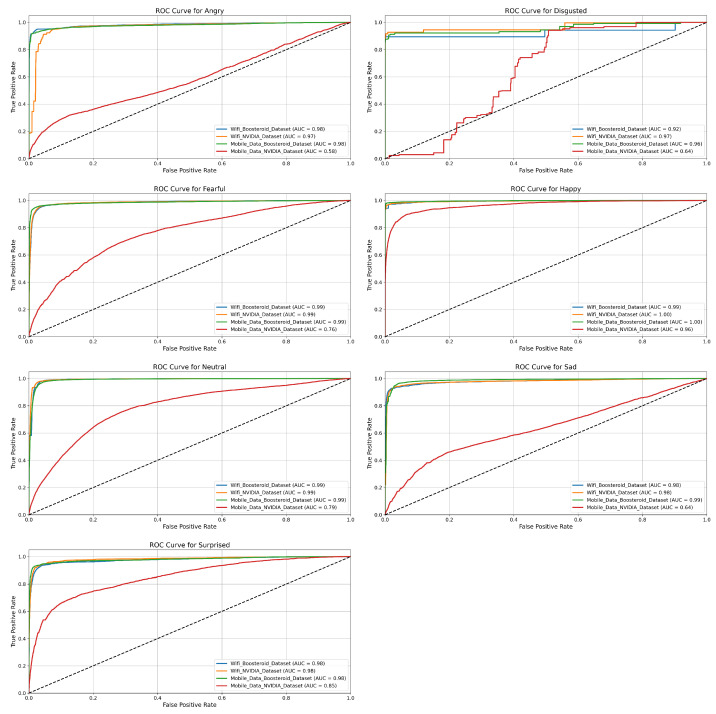
EmotionNET ROC of seven categories.

**Figure 5 sensors-25-01594-f005:**
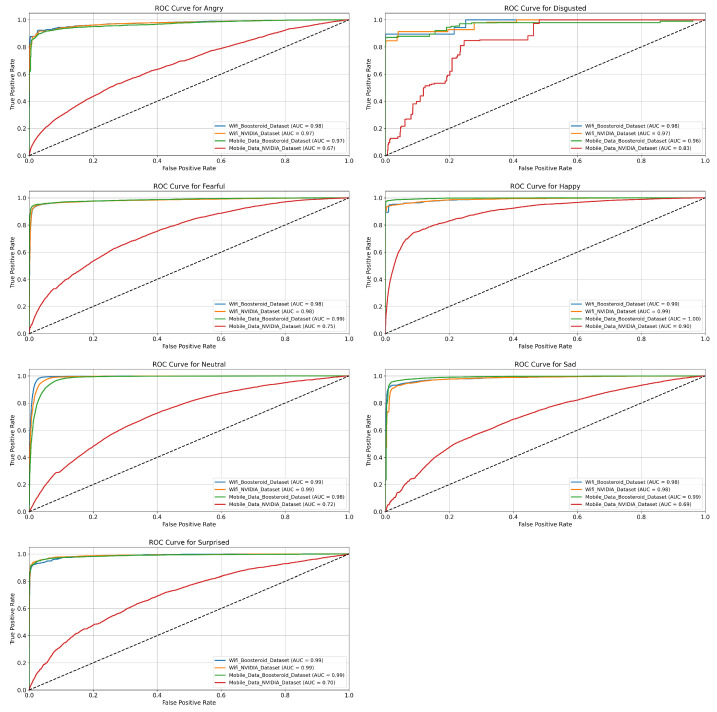
ConvoNEXT ROC of seven categories.

**Figure 6 sensors-25-01594-f006:**
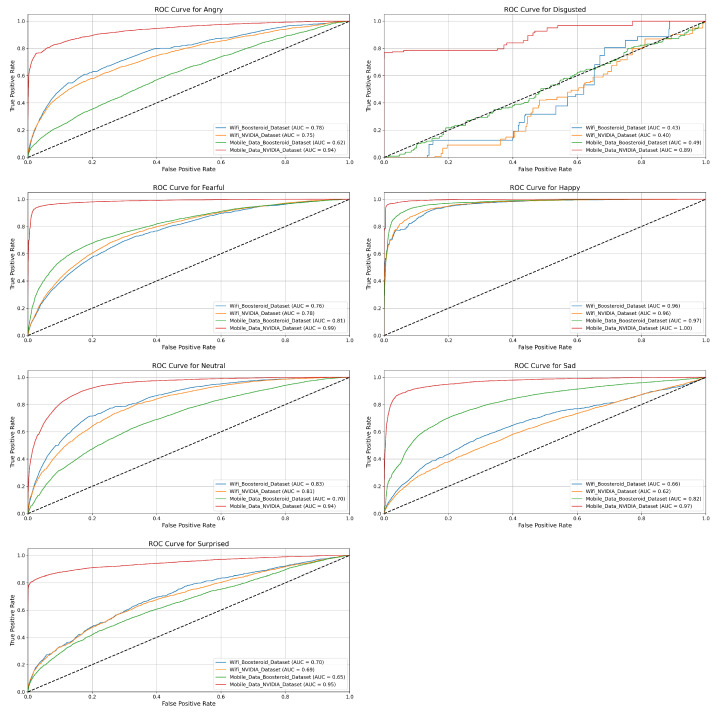
EfficientNET ROC of seven categories.

**Figure 7 sensors-25-01594-f007:**
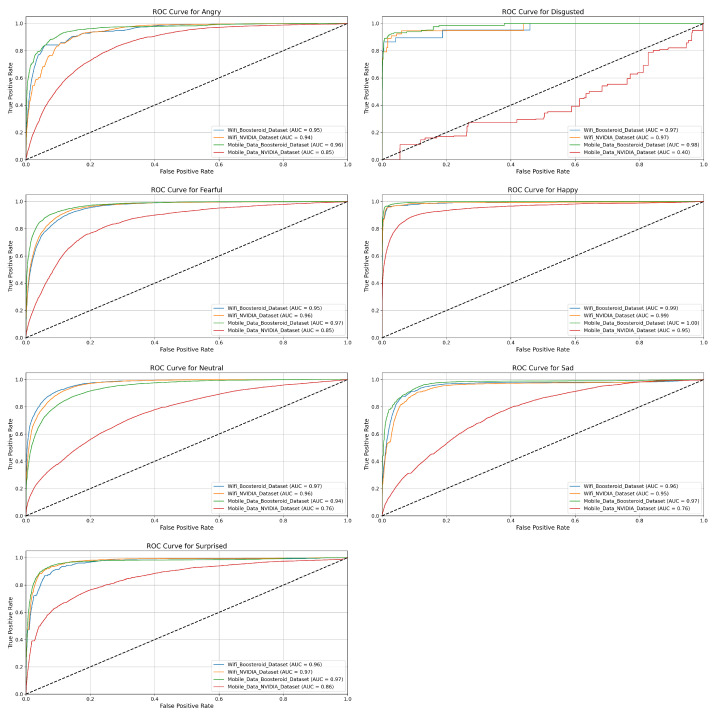
ViT ROC of seven categories.

**Figure 8 sensors-25-01594-f008:**
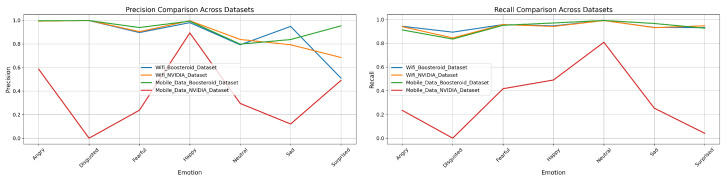
EmotionNET precision and recall.

**Figure 9 sensors-25-01594-f009:**
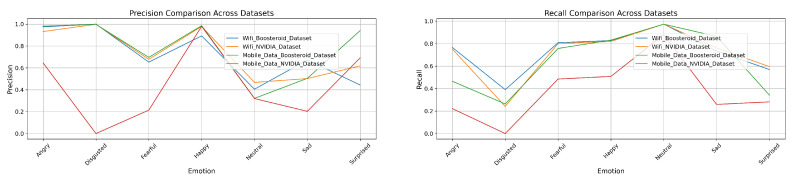
ConvoNEXT precision and recall.

**Figure 10 sensors-25-01594-f010:**
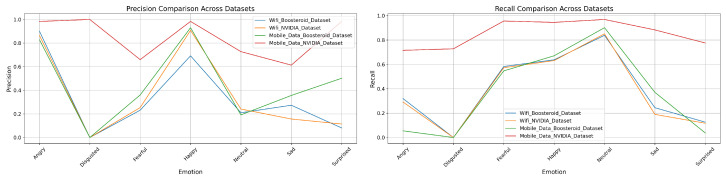
EfficientNET precision and recall.

**Figure 11 sensors-25-01594-f011:**
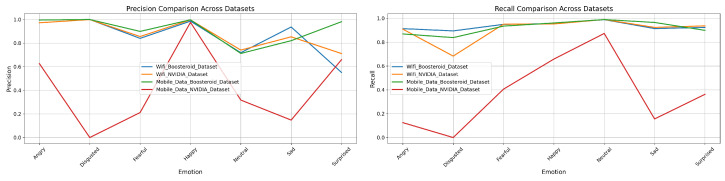
ViT precision and recall.

**Figure 12 sensors-25-01594-f012:**
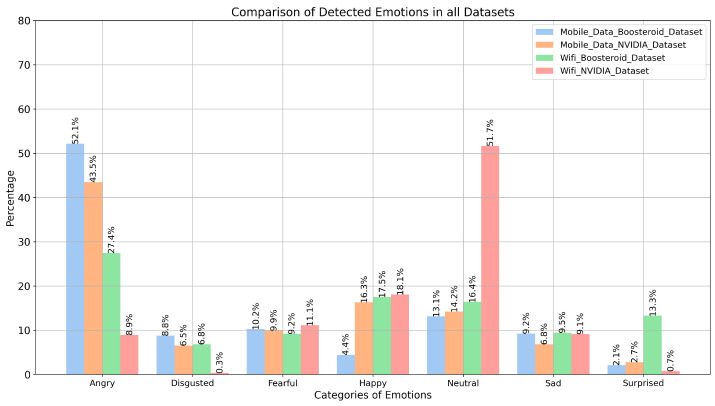
EmotionNET emotion comparisons.

**Table 1 sensors-25-01594-t001:** Comparison of different algorithms.

Algorithm/Technique Name	Model Architecture	Accuracy	Speed (FPS)	Training Dataset	Input Size	Pre-Trained Models	Implementation Complexity
DCM3-YOLOv4 [35]	CNN	94.3%	No	Derived from MAFA and WIDER FACE datasets	4000	No	Moderate
SSD [36]	ResNet v2, VGG16	Normal face recognition: 99.63% on LFW dataset, Occluded face recognition: 95.38% at 50% occlusion rate when 1 m away from the camera	8	MS1M-ArcFace dataset, CelebA dataset	85,000 and 5.8 million face images	Yes	Moderate
EfficientDet [37]	BiFPN	81.74%	No	COCO 2017	118K Images	Yes	Moderate
CenterNet [38]	CornerNet	84.5%	No	MS COCO	511 × 511	No	Moderate
Cascade R-CNN [39]	Parallel Cascade R-CNN	78.96%	No	DOTA (Dataset for Object Detection in Aerial Images)	800 × 800 to 4000 × 4000	Yes	Moderate
EmotionNET	CNN	98.9%	25	Facial Emotions (Custom Dataset)	48 × 48	No	Easy
ConvoNEXT	ResNet	94.9%	25	Facial Emotions (Custom Dataset)	48 × 48	No	Moderate
EfficientNET	EfficientNET-B0	92%	25	Facial Emotions (Custom Dataset)	48 × 48	No	Moderate
ViT	Transformer Encoder	91%	25	Facial Emotions (Custom Dataset)	48 × 48	No	Moderate

**Table 2 sensors-25-01594-t002:** Sample data distribution in WiFi and mobile data networks.

1	**WiFi Network**							**Image Size**
	**No. of Samples NVIDIA**							
	**Happy**	**Neutral**	**Fearful**	**Disgusted**	**Sad**	**Surprised**	**Angry**	
	358,570	205,405	68,966	71,028	47,349	95,447	243,236	
	**No. of Samples Boosteroid**							
	195,071	143,691	98,472	83,515	89,245	148,921	326,786	48*48
2	**Mobile Data Network**							**Image Size**
	**No. of Samples NVIDIA**							
	**Happy**	**Neutral**	**Fearful**	**Disgusted**	**Sad**	**Surprised**	**Angry**	
	183,853	128,088	96,884	82,177	61,708	21,177	489,447	
	**No. of Samples Boosteroid**							
	45,922	106,056	96,884	98,911	94,780	11,485	556,442	48*48

**Table 3 sensors-25-01594-t003:** Precision and recall of EmotionNET model with custom dataset.

Categories	WiFi			
	Boosteroid Dataset		NVIDIA Dataset	
	Precision	Recall	Precision	Recall
**Angry**	0.996451	0.943507	0.992839	0.940402
**Disgusted**	0.998307	0.894303	0.999209	0.845461
**Fearful**	0.896	0.958045	0.903822	0.95817
**Happy**	0.980498	0.946979	0.996376	0.942128
**Neutral**	0.792283	0.922221	0.837867	0.991881
**Sad**	0.948759	0.933745	0.79293	0.933245
**Surprised**	0.509869	0.932985	0.684749	0.947592
**Accuracy**	0.94432	0.94432	0.949001	0.949001
**Macro avg**	0.874681	0.943112	0.886828	0.936854
**Weighted Avg**	0.954899	0.94432	0.949317	0.949001
**Categories**	**Mobile Data**			
	**Boosteroid Dataset**		**NVIDIA Dataset**	
	**Precision**	**Recall**	**Precision**	**Recall**
**Angry**	0.997275	0.912834	0.585788	0.234193
**Disgusted**	0.997842	0.8358	0	0
**Fearful**	0.930996	0.952018	0.237341	0.417283
**Happy**	0.993108	0.97137	0.893458	0.490776
**Neutral**	0.797852	0.927797	0.294863	0.808724
**Sad**	0.837156	0.971615	0.120353	0.25301
**Surprised**	0.953285	0.927648	0.490402	0.014488
**Accuracy**	0.937588	0.937588	0.407133	0.407133
**Macro avg**	0.930802	0.937083	0.374598	0.320782
**Weighted Avg**	0.945943	0.937588	0.543855	0.407133

**Table 4 sensors-25-01594-t004:** Precision and recall of ConvoNEXT model technique with custom dataset.

Categories	WiFi			
	Boosteroid Dataset		NVIDIA Dataset	
	Precision	Recall	Precision	Recall
**Angry**	0.975075	0.76526	0.931996	0.7507
**Disgusted**	0.999612	0.390386	0.99824	0.241543
**Fearful**	0.652562	0.807772	0.678374	0.800201
**Happy**	0.892505	0.827109	0.979006	0.819233
**Neutral**	0.405371	0.972798	0.467218	0.972754
**Sad**	0.672179	0.733344	0.502351	0.761708
**Surprised**	0.44354	0.561911	0.618405	0.594901
**Accuracy**	0.752011	0.752011	0.75211	0.75211
**Macro avg**	0.72012	0.723694	0.73937	0.705863
**Weighted Avg**	0.848551	0.752011	0.834592	0.75211
**Categories**	**Mobile Data**			
	**Boosteroid Dataset**		**NVIDIA Dataset**	
	**Precision**	**Recall**	**Precision**	**Recall**
**Angry**	0.981519	0.464809	0.645119	0.221551
**Disgusted**	0.997433	0.26333	0	0
**Fearful**	0.696016	0.756588	0.214258	0.484776
**Happy**	0.985646	0.830706	0.97986	0.508122
**Neutral**	0.321687	0.972337	0.319671	0.879558
**Sad**	0.504303	0.863417	0.203163	0.259134
**Surprised**	0.940937	0.341246	0.619856	0.282032
**Accuracy**	0.622849	0.622849	0.448967	0.448967
**Macro avg**	0.775363	0.641776	0.436237	0.376453
**Weighted Avg**	0.826255	0.622849	0.609476	0.448967

**Table 5 sensors-25-01594-t005:** Precision and recall of EfficientNET model technique with custom dataset.

Categories	WiFi			
	Boosteroid Dataset		NVIDIA Dataset	
	Precision	Recall	Precision	Recall
**Angry**	0.900671	0.319717	0.864387	0.291317
**Disgusted**	0	0	0	0
**Fearful**	0.231446	0.582723	0.253531	0.573052
**Happy**	0.691953	0.637561	0.907626	0.63115
**Neutral**	0.209699	0.838441	0.241137	0.850652
**Sad**	0.273397	0.244553	0.15681	0.189921
**Surprised**	0.081096	0.126197	0.114437	0.117328
**Accuracy**	0.373277	0.373277	0.411646	0.411646
**Macro avg**	0.34118	0.392742	0.362561	0.37906
**Weighted Avg**	0.598176	0.373277	0.617448	0.411646
**Categories**	**Mobile Data**			
	**Boosteroid Dataset**		**NVIDIA Dataset**	
	**Precision**	**Recall**	**Precision**	**Recall**
**Angry**	0.828677	0.054913	0.983136	0.715405
**Disgusted**	0	0	0.999807	0.727721
**Fearful**	0.360429	0.546636	0.65981	0.955705
**Happy**	0.930001	0.670511	0.983295	0.944712
**Neutral**	0.19181	0.909071	0.727481	0.96838
**Sad**	0.356324	0.370756	0.613201	0.882893
**Surprised**	0.502004	0.037838	0.983005	0.775642
**Accuracy**	0.339924	0.339924	0.867067	0.867067
**Macro avg**	0.452749	0.368804	0.849462	0.852923
**Weighted Avg**	0.570115	0.339924	0.899558	0.867067

**Table 6 sensors-25-01594-t006:** Precision and recall of ViT model technique with custom dataset.

Categories	WiFi			
	Boosteroid Dataset		NVIDIA Dataset	
	Precision	Recall	Precision	Recall
**Angry**	0.995739	0.913334	0.97217	0.908918
**Disgusted**	0.999943	0.894303	0.999377	0.683195
**Fearful**	0.804108	0.949272	0.855533	0.50779
**Happy**	0.985952	0.955144	0.996585	0.952435
**Neutral**	0.718666	0.988827	0.74072	0.990046
**Sad**	0.936324	0.913752	0.853228	0.923594
**Surprised**	0.551169	0.923412	0.71139	0.936261
**Accuracy**	0.924894	0.924894	0.914069	0.914069
**Macro avg**	0.861129	0.934006	0.875554	0.904661
**Weighted Avg**	0.941075	0.924894	0.927702	0.914069
**Categories**	**Mobile Data**			
	**Boosteroid Dataset**		**NVIDIA Dataset**	
	**Precision**	**Recall**	**Precision**	**Recall**
**Angry**	0.995913	0.868784	0.625747	0.124897
**Disgusted**	0.999933	0.838172	0	0
**Fearful**	0.898994	0.934552	0.211776	0.405974
**Happy**	0.997525	0.960912	0.975668	0.658091
**Neutral**	0.712079	0.989735	0.317378	0.872986
**Sad**	0.820398	0.965266	0.147777	0.155881
**Surprised**	0.981273	0.898976	0.659422	0.362389
**Accuracy**	0.916666	0.916666	0.473058	0.473058
**Macro avg**	0.915159	0.922343	0.419684	0.36868
**Weighted Avg**	0.934024	0.916666	0.597963	0.473058

## Data Availability

The datasets and computational code supporting the findings of this study are not publicly available but can be obtained from the corresponding author upon reasonable request.

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
