# Peer review of "Quality of Experience (QoE) in Cloud Gaming: A Comparative Analysis of Deep Learning Techniques via Facial Emotions in a Virtual Reality Environment"

_sensors, 2025, doi:10.3390/s25051594_

Round 1
Reviewer 1 Report
Comments and Suggestions for Authors
This study proposed a deep learning based model to assess QoE. The findings showed that facial expressions are strongly correlated with QoE. While the proposed method has potential, I believe the manuscript needs to be substantially improved.
I don’t understand what the related work section is for. There is no clear link to the study’s contribution. It only shows superficial comparisons. I highly recommend putting together a combined section with the introduction, supporting your claims with evidence. You can explicitly identify the limitations of previous studies and explain how your study can resolve the issue.
Also, I suggest that the introduction acknowledge physiological sensors that can measure QoE or satisfaction, such as EEG, heart rate variability, and skin conductance, with specific examples. For instance, satisfaction during robotic hand control has been assessed using EEG sensors [1]. Since your study focuses on image data, it would be helpful to explicitly state this limitation early on.
[1] https://doi.org/10.3390/s23010277
The method section also lacks sufficient details.
There is no demographic information about the players like age, gaming experience.
Were they instructed to play in a specific way?
Reveal any inclusion or exclusion criteria.
There is no rationale for choosing the game. Also, you should introduce the game and what it has something useful for the experiment.
There are no camara specs.
How were the emotion labels assigned?
There is no training parameter information.
Provide all the details of performance metrics.
No statistical tests were mentioned, which is a significant flaw. This would make your results invalidated.
I don’t see real discussion whereas the section is named results and discussions.
I can’t find any references. You should interpret your results and compare them with previous studies and provide implications. I highly recommend separating the discussion from the results because the current description does not provide real discussion.
Author Response
Respected Reviewers
First of all, thank you so much for your precious time to evaluate our manuscript and help us to make it more good. I have attached our responses; please check. If you have any other queries, then please inform us.

Reviewer 2 Report
Comments and Suggestions for Authors
The research aims to evaluate cloud gaming Quality of Experience (QoE) using deep learning-based facial emotion recognition. It develops the EmotionNET model to analyze players' facial expressions, correlating them with network conditions to enhance real-time QoE assessment in gaming environments.
These comments should be answered carefully;
#How does the EmotionNET model perform in real-time cloud gaming environments with different latency and network conditions?
#To what extent does the dataset diversity impact the generalizability of EmotionNET in assessing QoE across various user groups?
#How do other network types, such as fiber-optic or satellite internet, influence QoE when assessed using deep learning-based emotion recognition?
#What measures can be implemented to reduce potential biases caused by individual and cultural differences in facial expression analysis for QoE assessment?
some points of the Related Work section need improvement, the authors should check them carefully;
# The section lists previous studies but does not critically analyze their limitations, weaknesses, or gaps.
#Although a comparison study is mentioned, it does not provide in-depth comparisons of methodologies, results, or effectiveness of different models.
#The section does not justify why the selected related works are the most relevant to the research problem.
#The discussion jumps between different topics (QoE, UGC videos, classification, etc.) without clear thematic organization.
#Some studies are described only at a high level without details on their methodologies or findings.
#The section does not explicitly highlight the shortcomings of previous studies, which is essential for justifying the current research.
#There is no clear linkage between the discussed related works and how the proposed research addresses gaps.
#While a table compares algorithms, there is no quantitative comparison of the related work in terms of performance, efficiency, or accuracy.
#Some referenced studies are mentioned without discussing how they contribute to the field or how their methodologies compare.
#The section lacks a summary that synthesizes key findings from related works and identifies a research gap to motivate the study.
In methodology, the authors should check these questions and clarify them why;
Is the sample size of 30 players and the limited use of a single game ("Fortnite") sufficient to provide a diverse and representative analysis of emotional responses, and can the findings be generalized to other games or a larger population?
Are the network conditions, such as Wi-Fi and 5G mobile data, adequately controlled for external factors like signal interference or variation in network performance, and could these factors introduce bias in the emotional response data?
#To strengthen the methodology and provide a more scientific approach, it is essential to include mathematical models that can formalize the relationships between the various parameters in the research study.
The authors should check these outcomes from obtained results;
# Discrepancy in Emotion Detection for "Fearful" and "Sad" Emotions, the "Fearful" emotion shows a slight increase in the Wifi Boosteroid dataset (11.1%) and a more consistent rate in the Wifi NVIDIA dataset (9.2%). This suggests the presence of fear-related emotions, but it is unclear why the fear component would increase while other negative emotions (like "Angry" and "Disgusted") decrease. Similarly, the "Sad" emotion remains fairly stable across both datasets (9.1%), which could be misrepresenting the emotional state of users. The low variation across datasets may need further clarification or improvement in dataset diversity and training conditions.
#The significant drop in "Surprised" emotion, especially in the Wifi NVIDIA dataset (0.7%), may be worth investigating further. This large variation suggests either a misclassification issue or the need for better training on detecting "Surprised" emotions, which could improve model accuracy.
#Underrepresentation of Negative Emotions, While positive emotions like "Happy" and "Neutral" show increases, negative emotions like "Angry," "Disgusted," and "Fearful" are less represented, particularly in the Wifi NVIDIA dataset. This imbalance could potentially limit the ability to assess full emotional spectrum shifts under different network conditions. More balanced datasets or better emotion classification might be required.
Comments on the Quality of English Language
The authors should provide English Proofreading.
Author Response
Respected Reviewers
Thank you so much for your precious time to evaluate our manuscript. We have addressed your concern to our manuscript, and we have tried to clear your doubts. If you feel any further queries, please let us know.

Round 2
Reviewer 1 Report
Comments and Suggestions for Authors
I don't have additional comments.
Reviewer 2 Report
Comments and Suggestions for Authors
The authors have implemented the corrections appropriately.
Comments on the Quality of English LanguageIt's preferable to proofread the draft.